# Sputum culture reversion in longer treatments with bedaquiline, delamanid, and repurposed drugs for drug-resistant tuberculosis

Sooyeon Kho[1], Kwonjune J. Seung[1,2], Helena Huerga[3], Mathieu Bastard[3], Palwasha Y. Khan[4,5], Carole D. Mitnick[1,2,6], Michael L. Rich[1,2], Shirajul Islam[7], Dali Zhizhilashvili[8], Lusine Yeghiazaryan[9], Elena Nikolaevna Nikolenko[10], Khin Zarli[11], Sana Adnan[12], Naseem Salahuddin[12], Saman Ahmed[13], Zully Haydee Ruíz Vargas[14], Amsalu Bekele[15], Aiman Shaimerdenova[16], Meseret Tamirat[17], Alain Gelin[18], Stalz Charles Vilbrun[19], Catherine Hewison[20,21], Uzma Khan[5,21] & Molly Franke[2,6,21] ✉

Sputum culture reversion after conversion is an indicator of tuberculosis (TB) treatment failure. We analyze data from the endTB multi-country prospective observational cohort (NCT03259269) to estimate the frequency (primary endpoint) among individuals receiving a longer (18-to-20 month) regimen for multidrug- or rifampicin-resistant (MDR/RR) TB who experienced culture conversion. We also conduct Cox proportional hazard regression analyses to identify factors associated with reversion, including comorbidities, previous treatment, cavitary disease at conversion, low body mass index (BMI) at conversion, time to conversion, and number of likely-effective drugs. Of 1,286 patients, 54 (4.2%) experienced reversion, a median of 173 days (97-306) after conversion. Cavitary disease, BMI < 18.5, hepatitis C, prior treatment with second-line drugs, and longer time to initial culture conversion were positively associated with reversion. Reversion was uncommon. Those with cavitary disease, low BMI, hepatitis C, prior treatment with second-line drugs, and in whom culture conversion is delayed may benefit from close monitoring following conversion.

In patients with tuberculosis disease (TB), sputum culture conversion from positive to negative serves as an interim indicator of treatment response[1-4], and correlates with the end of treatment outcome[5,6]. Culture reversion back to positive following conversion may portend treatment failure and identify patients who require close clinical follow-up, regimen adjustments, and/or adherence support[5,7-10]. Yet, few studies have quantified the frequency of reversion in cohorts of patients treated for multidrug- or rifampicin-

resistant (MDR/RR)-TB or identified factors that may influence its occurrence.

While the few existing analyses of culture reversion have been conducted among patients who received longer MDR/RR regimens, reversion is also relevant in the context of the shortened 6–9-month treatments endorsed by the World Health Organization (WHO) since May 2016[11,12]. Patient- and regimen-related factors that influence culture reversion in the context of longer regimens will likely serve to

identify subgroups who could benefit from a treatment duration that is longer than 6–9 months and/or those who should be closely followed for TB relapse after completion of a shortened regimen.

In this paper, we describe the frequency and rate of culture reversion among patients receiving longer regimens for MDR/RR-TB, mainly composed of bedaquiline, delamanid, linezolid, and clofazimine and identify patient- and regimen-related factors associated with culture reversion.

## Results

### Overview of cohort

Of 1286 patients who experienced conversion, the majority of patients (72.2%) had a history of prior treatment with second-line drugs (Table 1). Comorbidities were common—diabetes (17.8%), HIV infection (9.8%), hepatitis B infection (4.5%), and hepatitis C infection (11.9%). The median duration of treatment was 87 weeks (25th–75th percentile:

**Table 1 | Baseline characteristics of patients who experienced sputum culture conversion (N = 1286)**

| Characteristic | Results n (%)[a] |
|---|---|
| **At treatment initiation** | |
| Demographics | |
| Median age (years) (25th–75th percentile) | 35.0 (27.0-47.0) |
| Sex, Female | 440 (34.2) |
| Comorbidities | |
| Diabetes mellitus (N = 1273) | 227 (17.8) |
| HIV infection (N = 1284) | 126 (9.8) |
| Hepatitis B virus infection (N = 1282)[b] | 58 (4.5) |
| Hepatitis C virus infection (N = 1282)[c] | 152 (11.9) |
| TB-related characteristics | |
| Prior treatment with second-line drugs | 928 (72.2) |
| Resistance profile (N = 1284) | |
| MDR/RR without any injectable or fluoroquinolone resistance | 356 (27.7) |
| MDR/RR with any second-line injectable resistance | 146 (11.4) |
| MDR/RR with any fluoroquinolone resistance | 400 (31.2) |
| MDR/RR resistance to one injectable and one fluoroquinolone | 382 (29.8) |
| **At culture conversion** | |
| Median weeks to conversion (25th– 75th percentile) | 5.7 (4.1-9.1) |
| Body mass index < 18.5 (N = 1280) | 482 (37.7) |
| Number of likely effective drugs | |
| <4 | 250 (19.4) |
| ≥4 | 1036 (80.6) |
| Likely effective Group A drugs[d] | |
| 0 | 27 (2.1) |
| 1 | 256 (19.9) |
| 2 | 795 (61.8) |
| 3 | 208 (16.2) |
| Likely effective Group B drugs[e] | 198 (15.4) |
| 0 | 898 (69.8) |
| 1 | 190 (14.8) |
| 2 | |
| Cavitary disease (N = 1151) | 770 (66.9) |

TB tuberculosis, MDR multidrug-resistant, RR rifampin-resistant.
[a]Unless otherwise specified.
[b]Hepatitis B virus surface antigen positive.
[c]Hepatitis C virus antibody positive.
[d]Group A drugs: Fluoroquinolones (moxifloxacin or levofloxacin), bedaquiline, and linezolid.
[e]Group B drugs: Cycloserine or terizidone, and clofazimine.

84–101). Median time to conversion was 5.7 weeks (25th–75th percentile: 4.1–9.1). Median number of sputum cultures after conversion was 15 (25th–75th percentile: 9–18). At the time of culture conversion, 37.7% had a low BMI, and the median number of likely effective drugs was four. More than half (61.8%) of the patients had two likely effective group A drugs in their treatment regimen, and the majority were taking one or more likely effective group B drugs (84.6%). The majority (76.0%) of patients were taking bedaquiline, 39.0% were taking delamanid, and 17.0% were taking both bedaquiline and delamanid.

### Frequency and rate of reversion

54 (4.2%) patients experienced reversion during a total of 20,727 months of follow-up, a rate of 31.26 cases/1000 person-years (95% CI: 23.49–40.79). Among those who reverted, the median time to reversion was 173 days (25th–75th percentile: 97–306) from the time of conversion and 278 days (25th–75th percentile: 206–419) from treatment start.

### Patient-level factors associated with culture reversion

Table 2 shows the patient characteristics associated with reversion. Univariable Cox proportional hazards models show that low BMI at conversion (HR: 2.25, 95% CI: 1.31–3.87, p = 0.003), cavitary disease at conversion (HR: 3.29, 95% CI: 1.48–7.30, p = 0.003), hepatitis C infection (HR: 2.03, 95% CI: 1.04–3.93, p = 0.04) and previous TB treatment with second-line drugs (HR: 8.60, 95% CI: 1.19–62.27, p = 0.03) were associated with increased risk of reversion. In multivariable analyses, low BMI at conversion (aHR: 2.27, 95% CI: 1.30–3.96, p = 0.004), cavitary disease at conversion (aHR: 2.39, 95% CI: 1.07–5.36, p = 0.03), prior TB treatment with second-line drugs (aHR: 7.17, 95% CI: 0.98–52.49, p = 0.05), and hepatitis C infection at baseline (aHR: 2.18, 95% CI: 1.12–4.24, p = 0.02) were associated with increased risk of reversion. The variables for which each regression model was adjusted depended on the factor of interest and are footnoted in Table 2. Hepatitis B infection was excluded from the multivariable model because only one individual with baseline hepatitis B infection experienced reversion. Adjustment for substance use did not materially change the estimate for hepatitis C infection (aHR: 2.55, 95% CI: 1.28–5.09, p = 0.008). We found no change in effect estimates in sensitivity analyses censoring individuals with no post-conversion follow-up culture. Results from complete cases analyses and analyses that accounted for clustering by country were consistent with primary analyses.

### Treatment-related characteristics

Table 3 shows effect estimates for treatment-related characteristics. Both univariable and multivariable Cox proportional hazards model results show that longer time to conversion (HR: 1.05, 95% CI: 1.02–1.07, p = 0.001; aHR: 1.05, 95% CI: 1.02–1.08, p = 0.001) was associated with an elevated risk of reversion. The number of effective drugs (i.e., number of Group A drugs, number of Group B drugs, total number) was not associated with time to reversion. In sensitivity analyses in which we adjusted for patient-level factors associated with reversion (hepatitis C infection, cavitary disease, and low BMI), we found no substantive change in the effect estimates. Sensitivity analyses censoring individuals with no post-conversion follow-up culture yielded no substantive changes in the effect estimates.

### End-of-treatment outcomes

Among all patients who experienced conversion, the frequency of unsuccessful end-of-treatment outcomes was 17.6% (n = 226), with the majority of patients (82.4%; n = 1055) experiencing successful end-of-treatment outcomes. In contrast, among patients who experienced culture reversion, the majority (92.6%; n = 50) experienced unsuccessful end-of-treatment outcomes, including death (5.6%; n = 3), treatment failure (83.3%; n = 45), and loss to follow-up (3.7%; n = 2). 7.4% (n = 4) experienced successful end-of-treatment outcomes.

**Table 2 | Patient-level factors associated with reversion (N = 1286)[a]**

| | Reversion frequency n/N (%) | Univariable hazard ratio (95% CI)[b] | p value | Adjusted hazard ratio (95% CI) | p value |
|---|---|---|---|---|---|
| **At baseline** | | | | | |
| Diabetes mellitus[c] (N = 1273) | | | | | |
| No | 47/999 (4.7) | Reference | | Reference | 0.18 |
| Yes | 6/221 (2.7) | 0.57 (0.25–1.34) | 0.20 | 0.56 (0.24–1.31)[d] | |
| HIV infection | | | | | |
| No | 52/1106 (4.7) | Reference | | Reference | |
| Yes | 2/124 (1.6) | 0.39 (0.09–1.58) | 0.19 | 0.25 (0.05–1.20)[d] | 0.08 |
| Hepatitis B infection | | | | | |
| No | 53/1171 (4.5) | Reference | | | |
| Yes | 1/57 (1.8) | 0.39 (0.05–2.80) | 0.35 | | |
| Hepatitis C infection | | | | | |
| No | 43/1087 (4.0) | Reference | | Reference | |
| Yes | 11/141 (7.8) | 2.03 (1.04–3.93) | 0.04 | 2.18 (1.12–4.24)[d] | 0.02 |
| Previous TB Treatment | | | | | |
| None | 1/162 (0.6) | Reference | | Reference | |
| First-line drugs only | 3/192 (1.6) | 2.45 (0.25–23.56) | 0.44 | 2.38 (0.25–22.94)[e] | 0.45 |
| Second-line drugs | 50/878 (5.7) | 8.60 (1.19–62.27) | 0.03 | 7.17 (0.98–52.49)[e] | 0.05 |
| **At conversion** | | | | | |
| Cavitary disease[c] (N = 1151) | | | | | |
| No | 7/374 (1.9) | Reference | | Reference | |
| Yes | 45/725 (6.2) | 3.29 (1.48–7.30) | 0.003 | 2.39 (1.07–5.36)[f] | 0.03 |
| BMI < 18.5 | | | | | |
| No | 23/775 (3.0) | Reference | | Reference | |
| Yes | 30/452 (6.6) | 2.25 (1.31–3.87) | 0.003 | 2.27 (1.30–3.96)[f] | 0.004 |

[a]All tests are two-sided. No adjustments were made for multiple comparisons.
[b]CI: confidence interval.
[c]Univariable and multivariable models included missing indicator variables as needed.
[d]Model included diabetes mellitus, missing indicator for diabetes, HIV infection, and hepatitis C infection.
[e]Model included diabetes mellitus, missing indicator for diabetes, HIV infection, hepatitis C infection, and previous TB treatment at baseline.
[f]Model included diabetes mellitus, missing indicator for diabetes, HIV infection, hepatitis C infection, previous TB treatment at baseline, cavitary disease, missing indicator for cavitary disease, and low BMI at conversion.

**Table 3 | Treatment-related factors associated with reversion (N = 1286)[a]**

| | Reversion frequency n/N (%)[b] | Univariable hazard ratio (95% CI)[c] | p value | Adjusted[d] hazard ratio (95% CI) | p value |
|---|---|---|---|---|---|
| Time to conversion (weeks)[e] (median ± IQR) | 5.7 ± 5.0 | 1.05 (1.02–1.07) | 0.001 | 1.05 (1.02–1.08) | 0.001 |
| Number of effective drugs | | | | | |
| <4 | 15/235(6.4) | Reference | 0.10 | Reference | 0.53 |
| ≥4 | 39/997 (3.9) | 0.61 (0.33–1.10) | | 0.80 (0.39–1.62) | |
| Group A drugs | | | | | |
| 0 or 1 | 14/269(5.2) | 1.19 (0.51–2.75) | 0.69 | 1.25 (0.48–3.29) | 0.65 |
| 2 | 31/764 (4.1) | 0.90 (0.43–1.89) | 0.78 | 0.80 (0.37–1.72) | 0.57 |
| 3 | 9/199 (4.5) | Reference | | Reference | |
| Group B drugs | | | | | |
| 0 | 8/190 (4.2) | 7.76 (0.97–62.02) | 0.05 | 5.53 (0.67–45.98) | 0.11 |
| 1 | 45/853 (5.3) | 9.29 (1.28–67.42) | 0.03 | 6.27 (0.85–46.20) | 0.07 |
| 2 | 1/189 (0.5) | Reference | | Reference | |

[a]All tests are two-sided. No adjustments were made for multiple comparisons.
[b]Unless otherwise specified.
[c]CI: confidence interval.
[d]Model included time to conversion (weeks), number of effective drugs, group A drugs, group B drugs, and treatment site of Kazakhstan, which had the largest number of individuals who reverted (49.1%).
[e]Median ± interquartile range for all patients regardless of reversion status, for patients who experienced reversion: 8.5 ± 8.4 months, for patients who did not experience reversion: 5.5 ± 5.1 months.

## Discussion

Sputum culture reversion was rare among patients with MDR/RR-TB who were treated with bedaquiline- and/or delamanid-containing regimens. The low frequency of reversion (4.2%) compared favorably to previous studies, which have reported frequencies ranging from 5.4% to 22.7%[13–16]. While few studies have examined culture reversion during MDR/RR-TB treatment with drugs such as bedaquiline, delamanid, linezolid and clofazimine, our findings align with those from a retrospective cohort study conducted in South Africa, which found a lower rate of reversion among patients treated with bedaquiline, as compared to those who did not receive the drug[16]. Notably, in this present cohort, the median time to reversion among those who experienced it was 278 days from treatment initiation. This means that in at least half of those with reversion, it occurred at, or prior to, 9 months, the recommended duration of many shortened all oral treatments.

Regimens in this cohort mostly comprised four or more likely effective drugs, two or more group A drugs and at least one group B drug. The frequent use of high-quality regimens, together with the small number of reversion events, may explain the limited evidence of associations between regimen characteristics (i.e., more Group A and B drugs) and culture reversion. Instead, factors associated with advanced disease (low BMI and cavitary disease), history of previous TB treatment with second-line drugs, and a mostly untreated comorbid condition (hepatitis C infection) appeared to be more important determinants of reversion. The factors associated with advanced disease are known to delay culture conversion and reduce the likelihood of treatment success[17–19]. History of second-line TB treatment has also been associated with other unfavorable treatment outcomes, including death and recurrence[20–22]. The finding that hepatitis C infection is associated with the risk of reversion highlights a critical area for intervention: hepatitis C is common in many high-burden TB settings, and yet routine, systematic testing and treatment for hepatitis C is rare in patients receiving treatment for MDR/RR-TB[23]. An observational study conducted in Armenia demonstrated that it is possible to effectively treat chronic HCV infection in MDR-TB patients without major safety issues[24].

Time to sputum culture conversion is an important interim endpoint in TB treatment studies; at the individual level, it correlates well with end-of-treatment outcomes[6]. Our findings reaffirm that a longer time to conversion is an important marker of reversion risk[10] and suggest that individuals who experience a longer time to conversion represent a group who, at a minimum, require close monitoring but may also require longer or more intensive treatment. In addition, most patients (83.3%) who experienced reversion had an end-of-treatment outcome of failure. This reaffirms that reversion is an important warning sign for treatment failure.

A primary limitation to our analysis, the small number of reversions, which limited statistical power for analyzing determinants, reflects the extraordinary promise of regimens mainly composed of bedaquiline, delamanid, linezolid, and clofazimine. Relatedly, the large number of different regimen drug combinations received by patients in this cohort limited our ability to conduct a detailed analysis of the role of regimen composition on reversion[25]. And, while treatment changes between the time of initiation and culture conversion were rare, some regimens may have been changed following conversion, and these changes were not considered in this analysis. Another limitation of this analysis is the absence of data on reasons for missing cultures during follow-up. We chose not to censor at the time of the last culture because it is well known that individuals may be unable to produce a sputum sample as their clinical status improves. That results from a sensitivity analysis in which we removed the small number of individuals without follow-up cultures yielded similar findings to primary analyses is reassuring. Finally, while we built regression models based on the likely confounders for each specific factor of interest, residual or unmeasured confounding may prevent a causal interpretation of our estimates, and we were unable to analyze two key factors that may influence reversion: suboptimal treatment adherence[10,26] and acquired or treatment-emergent drug resistance[27,28].

In conclusion, culture reversion was infrequent among patients treated with longer regimens containing new and repurposed drugs. Indicators of advanced disease or treatment complications and other factors that contribute to delayed culture conversion may serve as markers for individuals who require close follow-up, particularly as shortened regimens become the prevailing standard of care.

## Methods

This study was approved by the Partners Healthcare Human Research Committee (Boston, MA, USA), the Médecins Sans Frontières Ethics Review Board (Geneva, Switzerland), IRD Institutional Review Board (Karachi, Pakistan) and in all 17 countries by local ethics committees or IRBs (Armenia: Ethics Committee of Yerevan State Medical University after Mkhitar Heratsi; Bangladesh: Ethical Committee, National Institute of Diseases of the Chest and Hospital; Belarus: Ethics Committees of the Republican Scientific and Practical Centre of Pulmonology and Tuberculosis; DPRK: Ministry of Public Health; Ethiopia: National Research Ethics Review Committee of Ministry of Scient and Technology; Georgia: Ethics Committee of National Center for Tuberculosis and Lung Diseases; Haiti: Comité Des Droits Humains Des Centres GHESKIO, Zanmi Lasante Research Committee; Indonesia: Ethics Committee, Faculty of Medicine, Universitas Indonesia; Kazakhstan: National Scientific Center of Phthisiopulmonology of the Ministry of Health; Kenya: The Scientific and Ethics Review Unit, Kenya Medical Research Institute; Kyrgyzstan: Committee on Bioethics under the MoH of the Kyrgyz Republic; Lesotho: Ministry of Health Research and Ethics Committee; Myanmar: Ethics Review Committee, Department of Medical Research, Ministry of Health and Sports; Pakistan: IRD Institutional Review Board; Peru: Institutional Research Ethics Committee at the Peruvian University of Cayetano Heredia; South Africa: Bio Medical Research Ethics Committee, University of KwaZulu-Natal; Vietnam: Sciences and Ethical Committee of the National Lung Hospital and Independent Ethics Committee, Ministry of Health). Informed consent was obtained by all participants prior to participation.

### Study design and patient population

Patients were enrolled in the endTB Observational Study (NCT03259269), a multi-center prospective cohort study of patients who were treated for MDR/RR-TB with a longer individualized regimen containing bedaquiline or delamanid in one of 17 participating countries. The primary efficacy (end-of-treatment outcomes) and safety (adverse events of clinical relevance) outcomes of this observational study have been published[25,29]. Commonly used repurposed drugs included linezolid and clofazimine. The composition of regimens was informed by the endTB clinical guide in accordance with local and WHO treatment guidelines. A common study protocol guided data collection, and data were entered into a single electronic medical record[30]. Patients included in this analysis initiated treatment between 1 April 2015 and 30 September 2018, had a positive baseline sputum culture, and experienced sputum culture conversion. Patients treated in the Democratic People's Republic of Korea, where comorbidity screening and follow-up procedures differed, were excluded from the analysis. Patients received directly observed therapy either in person or virtually in accordance with National Tuberculosis Program guidelines. As part of routine patient care, monthly sputum samples were collected for smears and cultures. Depending on local laboratory capacity and norms, cultures were grown in liquid (Mycobacterial Growth Indicator Tube) or solid medium, with many countries using both. 46.6% of patients were enrolled in countries using predominantly liquid cultures, and 53.4% were enrolled in countries using mainly solid media.

## Definitions

A positive baseline sputum culture was defined as any positive culture on a sputum specimen collected as early as 90 days before treatment initiation[31,32]. Culture conversion was defined as two consecutive negative cultures after treatment initiation collected at least 15 days apart[17]. Time to culture conversion was the time between the start of treatment and the first of the two negative cultures defining conversion. Among individuals who experienced conversion, culture reversion was defined as the presence of two positive cultures at least 30 days apart at any time after culture conversion[33]. Time to reversion was the time between the date of the second negative culture defining conversion and the first of the two positive cultures defining reversion. Follow-up time for those who did not experience reversion was defined as the time between the date of the second negative culture-defining conversion and the end of treatment. A low body mass index (BMI) was <18.5. Hepatitis B infection was defined as surface antigen positive, and hepatitis C infection was defined as antibody positive[17]. A drug was considered likely effective if all reported phenotypic or genotypic testing to that drug confirmed susceptibility, or if no resistance to the drug was reported and the patient had not previously received the drug for one month or more[17]. End-of-treatment outcome definitions were calculated to identify treatment failure at its earliest possible occurrence[34], based on 2013 World Health Organization outcomes definitions in place during the study period[33].

## Factors associated with reversion

Patient-level characteristics were assessed at the time of treatment initiation (diabetes, HIV infection, hepatitis B infection, hepatitis C infection, prior tuberculosis treatment history, and drug resistance profile) and at the time of conversion (BMI, presence of cavitation on chest radiograph). Since individuals can only experience sputum culture reversion after initial conversion, the most relevant period for assessing clinical characteristics that are affected by treatment (e.g., BMI, cavitation) is the time of conversion.

Regimen characteristics were assessed at the time of conversion and included time to conversion, the total number of likely-effective drugs, and a number of likely effective Group A and B drugs. Group A drugs included fluoroquinolones (moxifloxacin or levofloxacin), bedaquiline, and linezolid. Group B drugs consisted of cycloserine or terizidone and clofazimine. The number of Group C drugs, including delamanid, was not analyzed. Because drug resistance patterns would be expected to impact treatment outcomes through more severe disease (i.e., as indicated by the presence of cavities on chest radiograph or low BMI) or a less potent treatment regimen, we studied these more proximal factors rather than resistance pattern.

## Data analysis

The primary outcome was culture reversion. We calculated the frequency and rate of reversion and conducted univariable Cox regression analyses to identify factors associated with time-to-culture reversion. We confirmed that proportional hazard assumptions were held by testing whether the slope of Schoenfeld residuals from Cox models differed from zero, calculating covariate-specific and global p-values. Multivariable models were constructed based on potential confounders for each factor of interest, with patient characteristics considered separately from treatment characteristics because the latter may mediate an effect of the former. For the same reason, estimates for baseline characteristics (comorbidities and prior TB treatment) were not adjusted for factors assessed at the time of conversion (e.g., BMI and cavitary disease), and comorbidities were not adjusted for prior TB treatment. For variables with missing values, the missing indicator method was used. Patients with no sputum cultures after initial culture conversion were considered not to have reverted. We conducted sensitivity analyses in which we (1) ran complete case analyses in lieu of missing indicator analyses to address missing data; (2) adjusted for clustering by country using a Cox frailty model; (3)

censored the 23 individuals with no follow-up sputum cultures within 60 days from conversion; and (4) adjusted the model with treatment characteristics for patient-level correlates of reversion (hepatitis C infection, cavitary disease, and low BMI). For the analysis of hepatitis C infection, an additional sensitivity analysis was conducted adjusting for substance use, as it could be a potential confounder. Univariable and multivariable Cox regression analyses were performed using open-source code (survival package, v. 3.2-7) in R version 4.1.2.

## Reporting summary

Further information on research design is available in the Nature Portfolio Reporting Summary linked to this article.

## Data availability

Some of the data included in this analysis are managed in countries governed by the European Union General Data Protection Regulation (GDPR). The data contain sensitive and potentially identifying information and cannot be sufficiently anonymized to meet GDPR standards and retain their utility. Pseudoanonymized data will be made available within 2 weeks upon request to an MSF Medical Director at endTB.ClinicalTrial@paris.msf.org, and execution of a data sharing agreement or alternate means that allows assurance that principles of GDPR regulations will be met.

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

## Acknowledgements

The authors thank the patients who participated in the endTB Observational Study and the clinicians and program staff of participating national tuberculosis programs. They also thank endTB staff at Partners in Health, Doctors Without Borders, Epicentre, and Interactive Research and Development.

## Author contributions

S.K. led the data analysis and wrote the first draft of the paper; M.F. led the conceptualization, methodology, and contributed to the writing. C.D.M., M.F., M.L.R., K.J.S., U.K., P.Y.K., M.B., C.H. and H.H. led the design of the endTB study. U.K., C.H., M.B., S.I., D.Z., L.Y., E.N.N., K.Z., S.Ad., N.S., S.Ah., Z.H.R.V., A.B., A.S., M.T., A.G. and S.C.V. contributed to data collection and curation. All authors contributed to data interpretation, editing and critical review of the paper.

## Competing interests

The authors declare no competing interests.

## Additional information

[1]Division of Global Health Equity, Brigham and Women's Hospital, Boston, MA, USA. [2]Partners in Health, 800 Boylston Street Suite 300, Boston, MA, USA. [3]Epicentre, 14-34 Avenue Jean Jaurès, Paris, France. [4]Department of Clinical Research, London School of Hygiene & Tropical Medicine, London, UK.

[5]Interactive Research and Development Global, Singapore, Singapore. [6]Department of Global Health and Social Medicine, Harvard Medical School, Boston, MA, USA. [7]Interactive Research and Development, Dhaka, Bangladesh. [8]Médecins sans Frontières, Tbilisi, Georgia. [9]National Center for Pulmonology, Yerevan, Armenia. [10]Republican Research and Practical Centre for Pulmonology and Tuberculosis, Minsk, Belarus. [11]Médecins sans Frontières, Yangon, Myanmar. [12]Indus Hospital and Health Network, Karachi, Pakistan. [13]Interactive Research and Development, Karachi, Pakistan. [14]Maria Auxiliadora Hospital, San Juan de Miraflores, Peru. [15]Department of Internal Medicine, Tikur Anbessa Specialized Hospital and Addis Ababa University, College of Health Sciences, Addis Ababa, Ethiopia. [16]Karaganda Regional Center of Phthisiopulmonology, Karaganda, Kazakhstan. [17]Partners in Health, Lesotho, Maseru, Lesotho. [18]Zanmi Lasante, Port-au-Prince, Haiti. [19]Haitian Group for the Study of Kaposi's Sarcoma and Opportunistic Infections (GHESKIO), Port-au-Prince, Haiti. [20]Medical Department, Médecins sans Frontières, Paris, France. [21]These authors contributed equally: Catherine Hewison, Uzma Khan, Molly Franke. ✉e-mail: molly_franke@hms.harvard.edu

