## [Peer Review File · Nature Communications]

REVIEWER COMMENTS

Reviewer #1 (Remarks to the Author):

The authors present programmatic outcomes from several locations on factors associated with culture reversion in people with rifampin resistant and multi drug resistant TB treated over a few years with bedaquiline and delamanid containing regimens. This is an important undertaking.

Abstract

In the abstract, there is no mention that the study is in people with rifampin and multidrug resistant TB. For additional comments on abstract see below as well.

Introduction

In the first sentence the authors state that sputum culture conversion from positive to negative serves as an indicator of treatment response for patients with rifampin and multidrug resistant TB. This implies it is not an indicator for treatment outcome in people with drug susceptible TB, which is not correct, so I would recommend rewording the sentence.

Methods

In the section titled "correlates of conversion" the paragraph states that BMI and the presence of cavitation on chest radiograph at the time of conversion were used as variables in the analysis. Most TB studies look at presence of cavitation on chest radiograph at time of diagnosis or the initiation of therapy as risk factor for poorer outcomes. The authors use presence of cavitation at the time of conversion rather than at initiation of therapy. Please provide an explanation of why they did not use cavitation at the time of treatment initiation. Also, it would be interesting to see how the BMI at the time of diagnosis and cavitation at the time of diagnosis correlate with the BMI and cavitation at time of conversion. Was there a big change in BMI and cavitation status from time of diagnosis to time of conversion?

Also, in the abstract it should be noted that BMI at the time of conversion and cavitation a time of conversion is used in the model, rather than the baseline values since most readers would assume it is the baseline that is being referred to.

Given that adherence to treatment is correlated with improved outcomes for TB, please provide information in the methods on whether patients received treatment via direct observation, or how adherence was assessed during the study.

Title notes repurposed drugs were used in this study. Please list which ones are considered repurposed drugs as many readers may not be aware what this refers to.

Results

The authors list the group A and Group B drugs in the table 1 notes. However, delamanid is not listed as either belonging to Group A or Group B. In which category does it belong?

Since we do not know which group of drugs it belongs in, did they look at it as a separate variable and was the use of delamanid associated with treatment outcome?

What was the median duration of treatment in the study? The title states that they are looking at sputum culture reversion in people with longer treatments with newer drugs for drug resistant TB, but we have not been given information on the duration of treatment.

Also, they do not present information on how many people received bedaquiline or delamanid or both.

In the results a Kaplan-Meier curve or a reverse Kaplan-Meier curve would be helpful to show time to reversion.

In table 2, several different models are used for the multivariate analysis. What was the rationale for having model cited in footnote C versus the model cited in footnote D? The only difference between the models was previous TB treatment.

Table 1 shows the various categories of resistance. Was resistance profile associated with treatment outcomes in any of the models?

Table 4 information could be put in the text and instead the supplemental tables could be combined and presented as table 4. This way important data remains in the main text of the paper.

Discussion is well written and may need some edits based on the above comments.

Reviewer #2 (Remarks to the Author):

Abstract

1. Study cohort on longer regimens (of 18-20 months) not mentioned
2. Objective was worded as “.. to identify factors predicting reversion.” – this was not an analysis understand prediction per se, so suggest this is reworded

Methods

3. What culture methods were used?
4. Why was the first negative culture used to define time to reversion for those that did not experience reversion?
5. Would a sensitivity analysis with censoring date of last negative culture be feasible?
6. End of treatment outcomes - Rx failure used 2013 WHO definition - could this have mean some reversions being missed, as censored too early?
7. Adjustment for country was not conducted (except for binary variable “treated in Democratic People’s Republic of Korea” vs not)? – limitation of what can be done, due to small number of events. Did the authors consider robust SEs?
8. For variables with missing values, the missing indicator method was used - can the authors conduct sensitivity analyses for this based on complete case

Results

9. Can the authors report the end of treatment outcomes for the total cohort and not just those who reverted
10. I would not be so strict with P-values re. interpretation of results- see adjusted analysis for previous TB treatment with 2nd line drugs
11. For analysis for cavitory disease and BMI - are the authors concerned about over parameterization of the model? I think each include 10 parameters (including missing value indicators) and only 55 events – for binary outcomes this large number of parameters would be problematic for the logistic model
12. Table 3- “time to conversion” was considered in its continuous form as a linear effect. Did the authors assess for more complex relationships (though recognize there are a small number of events)

13. What did the Schoenfeld residuals from Cox models reveal? - brief comment would be useful

Discussion

14. Rather than BMI at culture conversion, would it be of interest to look at BMI change from starting treatment to culture conversion?

15. Agree that having some measure of adherence would have been interesting - why was this not possible?

16. Was there any major changes in the treatment regimen in follow-up – though not suggesting you look at this time-dependent covariate, it would be useful to report such data.

Minor issues

17. Abstract: add “longer weeks to initial culture conversion were associated with an increased rate of reversion”

18. Methods: acronyms BDQ & DLM used without explanation

19. Define group A & B drugs in methods

20. Results: Check rate calculation for reversion $55/22,882 = 24.04$ (?not 28.84/1000 pyrs)

21. Results: Table 1 – add “in years” for median age

22. Results: Can you add the number of reversion events to Tables 2 and 3?

23. Discussion (last paragraph): do you mean “.. contribute to delayed culture conversion”

Reviewers' Comments to the Authors:

Reviewer #1 (Remarks to the Author):

The authors present programmatic outcomes from several locations on factors associated with culture reversion in people with rifampin resistant and multi drug resistant TB treated over a few years with bedaquiline and delamanid containing regimens. This is an important undertaking.

Abstract

In the abstract, there is no mention that the study is in people with rifampin and multidrug resistant TB. For additional comments on abstract see below as well.

Response: We have added this to the abstract on page 2, line 4.

Introduction

In the first sentence the authors state that sputum culture conversion from positive to negative serves as an indicator of treatment response for patients with rifampin and multidrug resistant TB. This implies it is not an indicator for treatment outcome in people with drug susceptible TB, which is not correct, so I would recommend rewording the sentence.

Response: Thank you for pointing this out. We have reworded the sentence in the introduction to clarify that this applies generally to patients undergoing treatment for TB.

Methods

In the section titled "correlates of conversion" the paragraph states that BMI and the presence of cavitation on chest radiograph at the time of conversion were used as variables in the analysis. Most TB studies look at presence of cavitation on chest radiograph at time of diagnosis or the initiation of therapy as risk factor for poorer outcomes. The authors use presence of cavitation at the time of conversion rather than at initiation of therapy. Please provide an explanation of why they did not use cavitation at the time of treatment initiation. Also, it would be interesting to see how the BMI at the time of diagnosis and cavitation at the time of diagnosis correlate with the BMI and cavitation at time of conversion. Was there a big change in BMI and cavitation status from time of diagnosis to time of conversion?

Response: Substantial changes in clinical variables, such as cavitory disease and low BMI may occur in the first few months of treatment. Since individuals can only experience sputum culture reversion after initial conversion, the most relevant period for assessing clinical characteristics that are affected by treatment (e.g., BMI, cavitation) is at the time of conversion. We have added this sentence to the manuscript on page 5, lines 20-22.

Also, in the abstract it should be noted that BMI at the time of conversion and cavitation a time of conversion is used in the model, rather than the baseline values since most readers would

assume it is the baseline that is being referred to.

Response: We have added this to the abstract on page 2, lines 7-8.

Given that adherence to treatment is correlated with improved outcomes for TB, please provide information in the methods on whether patients received treatment via direct observation, or how adherence was assessed during the study.

Response: The methods section has been updated, as reflected below on page 4, lines 12-24.

“Patients received directly observed therapy either in person or virtually in accordance with National Tuberculosis Program guidelines.”

Title notes repurposed drugs were used in this study. Please list which ones are considered repurposed drugs as many readers may not be aware what this refers to.

Response: The methods section has been updated to state that the commonly used repurposed drugs were linezolid and clofazimine, on page 4, lines 5-6.

Results

The authors list the group A and Group B drugs in the table 1 notes. However, delamanid is not listed as either belonging to Group A or Group B. In which category does it belong?

Response: Delamanid belongs to Group C, which was not assessed in this study. We have added a sentence indicating this on page 6, lines 4-5.

Since we do not know which group of drugs it belongs in, did they look at it as a separate variable and was the use of delamanid associated with treatment outcome?

Response: Thank you for raising this point. We focused on Group A and B drugs as they are endorsed by the WHO and the American Thoracic Society as priority drugs in the treatment of TB.

What was the median duration of treatment in the study? The title states that they are looking at sputum culture reversion in people with longer treatments with newer drugs for drug resistant TB, but we have not been given information on the duration of treatment.

Response: Thank you for pointing this out. The median duration of treatment was 87 weeks (25th-75th percentile: 84-101), and we have added this to the results section on page 7, line 12.

Also, they do not present information on how many people received bedaquiline or delamanid or both.

Response: We have updated the results section, as reflected below on page 7 lines 17-19.

“The majority (76.0%) of patients were taking bedaquiline, 39.0% were taking delamanid, and 17.0% were taking both bedaquiline and delamanid.”

In the results a Kaplan-Meier curve or a reverse Kaplan-Meier curve would be helpful to show time to reversion.

Response: Thank you for this suggestion. We have elected not to include a KM-curve and instead to convey this information via an incidence rate and 95% confidence intervals, mainly because the very small rate of conversion makes it challenging to scale the figure in such a way that it is informative and not prone to misinterpretation.

In table 2, several different models are used for the multivariate analysis. What was the rationale for having model cited in footnote C versus the model cited in footnote D? The only difference between the models was previous TB treatment.

Response: We selected model covariates based on confounders. In estimating associations for HIV, diabetes and hepatitis C, we elected not to adjust for previous treatment with second-line drugs because all three comorbidities may impact TB treatment outcomes (and therefore prior TB treatment may be on the causal pathway and not a confounder). We provide an explanation in the Data Analysis section of the paper.

Table 1 shows the various categories of resistance. Was resistance profile associated with treatment outcomes in any of the models?

Response: Thank you for raising an important point. Resistance profile would be expected to impact reversion through patient characteristics such as cavitory disease (i.e., a high degree of resistance may lead to more advanced disease, as manifested by cavitory disease or low BMI) – or through the quality of the regimen (i.e., a high degree of resistance may lead to fewer group A or B drugs in the regimen). For this reason, we focused on indicators of disease severity and quality of treatment, rather than resistance patterns. We added this information on page 6, lines 5-8.

Table 4 information could be put in the text and instead the supplemental tables could be combined and presented as table 4. This way important data remains in the main text of the paper.

Response: Thank you for this suggestion. The results section has been updated, such that Table 4 information is now in the text on page 9, lines 9-12. We have also combined the supplemental tables with Tables 2 and 3, and removed Table 4.

Discussion is well written and may need some edits based on the above comments.

Response: Thank you. We have made edits to the discussion.

Reviewer #2 (Remarks to the Author):

Abstract

1. Study cohort on longer regimens (of 18-20 months) not mentioned

Response: We have added this to the abstract.

2. Objective was worded as “.. to identify factors predicting reversion.” – this was not an analysis understand prediction per se, so suggest this is reworded

Response: We agree and have reworded the sentence in the introduction as reflected below on page 2, lines 9-10.

“Cox proportional hazard regression analyses were performed to identify factors associated with reversion.”

Methods

3. What culture methods were used?

Response: We have updated the methods section, as reflected below on page 4, lines 15-18.

“Depending on local laboratory capacity and norms, cultures were grown in liquid (Mycobacterial Growth Indicator Tube) or solid medium, with many countries using both. 46.6% of patients were enrolled in countries using predominantly liquid cultures, and 53.4% were enrolled in countries using mainly solid media”

4. Why was the first negative culture used to define time to reversion for those that did not experience reversion?

Response: Thank you for pointing this out. We corrected the text to say that the second negative culture was used to define follow-up time for those that did not experience reversion, as reflected on page 5, lines 6-7.

5. Would a sensitivity analysis with censoring date of last negative culture be feasible?

Response: We appreciate the spirit of this comment; however, because patients who are evolving well clinically often cannot produce a sputum sample, a sensitivity analysis in which we censored at the time of the last negative culture would under-estimate the person-time. Furthermore, this underestimation could be differential across characteristics of interest (i.e., because patients living with HIV often produce less sputum). We have added this as a limitation to the analysis.

6. End of treatment outcomes - Rx failure used 2013 WHO definition - could this have mean some reversions being missed, as censored too early?

Response: Anyone who failed treatment due to positive culture following culture conversion would have been classified as a case of reversion. We do not believe reversions could have been missed due to early censoring.

7. Adjustment for country was not conducted (except for binary variable “treated in Democratic People’s Republic of Korea” vs not)? – limitation of what can be done, due to small number of events. Did the authors consider robust SEs?

Response: In this revised version of the manuscript, we opted to exclude patients treated in DPRK. While the results are identical either way, the theoretical differences in follow-up, led us to the decision to exclude these participants. We reran final analyses using robust standard errors and results were consistent with primary analyses. We now note this in the paper.

8. For variables with missing values, the missing indicator method was used - can the authors conduct sensitivity analyses for this based on complete case

Response: Thank you for this suggestion. We conducted sensitivity analyses using complete case analyses, and have compiled the results into a separate table, as reflected below. Effect estimates were similar but less precise in the complete case analyses, and in some cases, estimates for potential confounders became unstable. We therefore opted not to show these results in the paper; however we mention the sensitivity analysis.

Supplemental Table 1. Missing Indicator and Complete Case Analyses of Patient-level Factors Associated with Reversion (N=1,286)

	Missing Indicator Analysis		Complete Case Analysis	
	Adjusted Hazard Ratio (95% CI) ^a	P value	Adjusted Hazard Ratio (95% CI)	P value
At Baseline				
Diabetes mellitus ^b (N=1,273)				
No	Reference		Reference	
Yes	0.56 (0.24-1.31) ^c	0.18	0.56 (0.24-1.32) ^d	0.19
HIV infection				
No	Reference		Reference	
Yes	0.25 (0.05-1.20) ^c	0.08	0.37 (0.09-1.53) ^d	0.17
Hepatitis C infection (N=1,319)				
No	Reference		Reference	
Yes	2.18 (1.12-4.24) ^c	0.02	2.15 (1.10-4.19) ^d	0.02
Previous TB Treatment (N=1,330)				
None	Reference		Reference	
First-line drugs only	2.38 (0.25-22.94) ^e	0.45	2.41 (0.25-23.21) ^f	0.45

Second-line drugs	7.17 (0.98-52.49) ^e	0.053	6.83 (0.94-49.90) ^f	0.058
At Conversion				
Cavitory disease ^b				
No	Reference		Reference	
Yes	2.39 (1.07-5.36) ^g	0.03	2.37 (1.05-5.32) ^h	0.04
BMI <18.5 ^b				
No	Reference		Reference	
Yes	2.27 (1.30-3.96) ^g	0.004	2.03 (1.15-3.59) ^h	0.02

^a CI: confidence interval

^b Univariable and multivariable models included missing indicator variables as needed

^c Model included diabetes mellitus, missing indicator for diabetes, HIV infection, and hepatitis C infection

^d Model included diabetes mellitus, HIV infection, and hepatitis C infection

^e Model included diabetes mellitus, missing indicator for diabetes, HIV infection, missing indicator for HIV, hepatitis C infection, and previous TB treatment at baseline

^f Model included diabetes mellitus, HIV infection, hepatitis C infection, and previous TB treatment at baseline

^g Model included diabetes mellitus, missing indicator for diabetes, HIV infection, hepatitis C infection, and previous TB treatment at baseline; cavitory disease, missing indicator for cavitory disease, and low BMI at conversion

^h Model included diabetes mellitus, HIV infection, hepatitis C infection, and previous TB treatment at baseline; cavitory disease and low BMI at conversion

Results

9. Can the authors report the end of treatment outcomes for the total cohort and not just those who reverted.

Response: Thank you for this suggestion. The results section has been updated, such that end-of-treatment outcomes for those who experienced conversion are included in the text, reflected below on page 9, lines 7-9.

“Among all patients who experienced conversion, the frequency of unsuccessful end-of-treatment outcomes was 17.6% (n=226), with the majority of patients (82.4%; n=1,055) experiencing successful end-of-treatment outcomes.”

10. I would not be so strict with P-values re. interpretation of results- see adjusted analysis for previous TB treatment with 2nd line drugs

Response: Thank you for this suggestion – we agree. The results and discussion section has been updated accordingly, reflected below on page 8, lines 7-11, and page 10, lines 11-12.

“In multivariable analyses, low BMI at conversion (aHR: 2.27, 95% CI: 1.30-3.96, p=0.004), cavitory disease at conversion (aHR: 2.39, 95% CI: 1.07-5.36, p=0.03), prior TB treatment with second-line drugs (aHR: 7.17, 95% CI: 0.98-52.49, p=0.05), and hepatitis C infection at baseline (aHR: 2.18, 95% CI: 1.12-4.24, p=0.02) were associated with increased risk of reversion.”

“History of second-line TB treatment has also been associated with other unfavorable treatment outcomes, including death and recurrence.²⁵⁻²⁷”

11. For analysis for cavitory disease and BMI - are the authors concerned about over parameterization of the model? I think each include 10 parameters (including missing value indicators) and only 55 events – for binary outcomes this large number of parameters would be problematic for the logistic model

Response: Thank you for raising this point. We recognize we are at the upper bound of what the model can accommodate with 54 of events (a maximum of 8 parameters, including the missing indicator variables). We are not concerned about overparameterization due to the relatively similar standard error values in both the univariable and multivariable analyses.

12. Table 3- “time to conversion” was considered in its continuous form as a linear effect. Did the authors assess for more complex relationships (though recognize there are a small number of events)

Response: We did not assess more complex relationships for two reasons. The first reason was that raised by the reviewer (i.e., the small number of events). Second, we believed that the relationship between time to conversion and risk of reversion was linear (i.e., the rate of reversion increases linearly with day without conversion).

13. What did the Schoenfeld residuals from Cox models reveal? - brief comment would be useful

Response: Thank you for this suggestion. To examine whether proportional hazards assumptions held, we tested whether the slope of Schoenfeld residuals differed from zero, calculating covariate-specific and global p-values. We have added this on page 6, lines 13-15.

Discussion

14. Rather than BMI at culture conversion, would it be of interest to look at BMI change from starting treatment to culture conversion?

Response: Thank you for raising this point. We studied BMI as a binary variable, and used a meaningful and well-established threshold of 18.5 to indicate severe malnutrition in patients. We agree it would be of interest to look at BMI change from treatment start to conversion; however, this would be more challenging to interpret and likely depends on one’s baseline BMI value (for example, a 1-unit BMI change from 16 to 17 may have a different impact than a BMI change from 26 to 27). A robust analysis of BMI change and reversion would require a more comprehensive and nuanced analysis that is beyond the scope of this paper.

15. Agree that having some measure of adherence would have been interesting - why was this not possible?

Response: While it would have been interesting to explore adherence, it is a limitation of this study. Although everyone in this study had DOT, the reliability of adherence data varied across study sites.

16. Was there any major changes in the treatment regimen in follow-up – though not suggesting you look at this time-dependent covariate, it would be useful to report such data.

Response: Treatment often did change over time in this cohort. Most frequent were planned changes, such as the suspension of the injectable or bedaquiline (after 6 months). On the other hand, there were very few changes in treatment from initiation to culture-conversion. We have added this to the discussion on page 11, lines 7-9.

Minor issues

17. Abstract: add “longer weeks to initial culture conversion were associated with an increased rate of reversion”

Response: We have made this change.

18. Methods: acronyms BDQ & DLM used without explanation

Response: Thank you for pointing this out. We have removed these abbreviations.

19. Define group A & B drugs in methods

Response: Thank you for this suggestion. We have added definitions for group A and B drugs in the methods section, as reflected below on page 6, lines 3-4.

“Group A drugs included fluoroquinolones (moxifloxacin or levofloxacin), bedaquiline, and linezolid. Group B drugs consisted of cycloserine or terizidone, and clofazimine.”

20. Results: Check rate calculation for reversion $55/22,882 = 24.04$ (?not 28.84/1000 pyrs)

Response: Our calculation is correct. $55/22,882=0.002404$ cases per month. We multiply this time 12 months and get 0.02884 cases per year. We multiply this by 1000 and get 28.84 cases per 1000 person years.

After excluding participants from the Democratic People’s Republic of Korea and defining follow-up time for those who did not experience reversion from the date of the second negative culture defining conversion, there are now 54 cases with 20,727 months of follow up. $54/20,727=0.002605$ cases per month, multiplied by 12 months and get 0.03126 cases per year. We multiply by 1000 and get 31.26 cases per 1000 person years.

21. Results: Table 1 – add “in years” for median age

Response: Thank you for this suggestion. We have fixed Table 1 on page 15 of the revised manuscript.

22. Results: Can you add the number of reversion events to Tables 2 and 3?

Response: We have added the number of reversion events to Tables 2 and 3.

23. Discussion (last paragraph): do you mean “.. contribute to delayed culture conversion”

Response: Thank you for pointing this out. We have fixed this in the discussion on page 11, line 21.

REVIEWERS' COMMENTS

Reviewer #1 (Remarks to the Author):

Thank you for addressing the comments. Only suggestion to to say in abstract in results that the cavitaion status and BMI were at time of conversion.

Reviewers' Comments to the Authors:

Reviewer #1 (Remarks to the Author):

Thank you for addressing the comments. Only suggestion to say in abstract in results that the cavitation status and BMI were at time of conversion.

Response: We have added this to the abstract.